# Real-time synthesis of imagined speech processes from minimally invasive recordings of neural activity

Miguel Angrick [1✉], Maarten C. Ottenhoff [2], Lorenz Diener[1], Darius Ivucic[1], Gabriel Ivucic[1], Sophocles Goulis[2], Jeremy Saal[2], Albert J. Colon[3], Louis Wagner[3], Dean J. Krusienski[4], Pieter L. Kubben[2,5], Tanja Schultz[1] & Christian Herff [2✉]

Speech neuroprosthetics aim to provide a natural communication channel to individuals who are unable to speak due to physical or neurological impairments. Real-time synthesis of acoustic speech directly from measured neural activity could enable natural conversations and notably improve quality of life, particularly for individuals who have severely limited means of communication. Recent advances in decoding approaches have led to high quality reconstructions of acoustic speech from invasively measured neural activity. However, most prior research utilizes data collected during open-loop experiments of articulated speech, which might not directly translate to imagined speech processes. Here, we present an approach that synthesizes audible speech in real-time for both imagined and whispered speech conditions. Using a participant implanted with stereotactic depth electrodes, we were able to reliably generate audible speech in real-time. The decoding models rely predominantly on frontal activity suggesting that speech processes have similar representations when vocalized, whispered, or imagined. While reconstructed audio is not yet intelligible, our real-time synthesis approach represents an essential step towards investigating how patients will learn to operate a closed-loop speech neuroprosthesis based on imagined speech.

[1] Cognitive Systems Lab, University of Bremen, Bremen, Germany. [2] Department of Neurosurgery, School of Mental Health and Neurosciences, Maastricht University, Maastricht, The Netherlands. [3] Academic Center for Epileptology, Kempenhaeghe/Maastricht University Medical Center, Kempenhaeghe, The Netherlands. [4] ASPEN Lab, Biomedical Engineering Department, Virginia Commonwealth University, Richmond, VA, USA. [5] Academic Center for Epileptology, Kempenhaeghe/Maastricht University Medical Center, Maastricht, The Netherlands. ✉email: miguel.angrick@uni-bremen.de; c.herff@maastrichtuniversity.nl

Steady progress has been made in the field of brain–computer interfaces in recent years[1] and systems allow paralyzed patients to control robotic arms[2] or computer cursors[3] with high reliability and have been tested and used by patients in their daily lives[4,5]. While such systems can greatly increase the independence of disabled patients, they are currently unable to provide a natural means of communication for patients who are unable to speak. Recently, several studies have demonstrated the ability to synthesize high-fidelity audio directly from brain activity during audibly vocalized[6,7] and silently mimed speech[8]. These approaches are based on neural activity measured directly on the cortical surface using electrocorticographic (ECoG) electrode arrays. Studies employing intracortical microarrays have presented similar success in the decoding of speech processes[9,10].

Despite the impressive reconstruction quality achieved by these approaches, two major issues remain unaddressed in the current literature: (1) all studies synthesizing speech from the neural activity are based on open-loop experiments and apply the decoding pipelines to these data in offline analyses. In particular, decoding models based on artificial neural networks[6,8,11] require multiples of real-time processing for reliable decoding. (2) Recent advances in speech decoding have been demonstrated on audibly vocalized or silently mimed speech by individuals that have the ability to speak. To reach the target population of patients that are unable to speak, it is imperative to demonstrate efficacy in decoding imagined or attempted speech.

These two issues have been investigated in separate studies. Two studies by Moses et al.[12,13] decoded neural activity into a textual representation in real-time, but rely on produced or perceived speech. Two studies by Martin et al.[14,15] examine imagined speech processes but are based on offline processing of open-loop datasets. To date, the only research addressing both issues is the landmark study that synthesized vowel sounds from a patient with a neurotrophic electrode implanted into the motor cortex[16]. However, this study did not attempt to synthesize speech beyond basic vowel sounds.

Here, we show the feasibility of real-time synthesis of imagined spoken words from neural recordings, thereby directly addressing the two open challenges in the development of speech neuroprosthetics.

**Closed-loop synthesis.** In this study with one participant (20 years old, female), we recorded intracranial neural activity during speech processes using stereotactic electroencephalography (sEEG) electrodes. The use of sEEG electrodes has several advantages over other intracranial recording modalities, including reduced surgical trauma and implications for long-term implantation[17].

Our experiment consisted of two stages: an open-loop stage for training the decoding models, followed by a closed-loop stage where whispered and imagined speech were respectively evaluated (Fig. 1). In this context, an open-loop experiment refers to the collection of data without immediate feedback provided by the neuroprosthesis. In a closed-loop experiment, the neural data is analyzed and decoded in real-time, and feedback is provided by the neuroprosthesis. The neuroprosthesis is thereby closing the loop.

For the open-loop stage, the participant was presented with a single word on a monitor and instructed to speak the word aloud. The intracranial neural activity and acoustic speech signal were simultaneously recorded for a series of different words. These synchronized data were used to train the decoding models for the estimation of the acoustic speech waveform for the subsequent closed-loop stage.

For the closed-loop stage, real-time acoustic feedback was provided to the participant through our real-time synthesis approach (Fig. 2), which was trained using data from the open-loop stage. During the first closed-loop run, the participant was instructed to produce whispered speech. For the second closed-loop run, the participant was instructed to imagine producing the prompted words without vocalizing or activating speech articulators. As feedback was provided in real-time, any potential acoustic contamination of the neural signals[18] will not contribute to the decoding performance.

Each run consisted of 100 Dutch words, which were displayed for two seconds followed by a fixation cross for one second. Words were drawn randomly from a phonetically balanced list of 250 words[19].

**Decoding approach.** Our approach receives raw sEEG signals and performs a real-time conversion to an acoustic speech signal, which is provided as continuous auditory feedback to the user. In this design, an auditory waveform is continuously produced, independent of explicit detection of speech or silence onset. When functioning as intended, the system will provide audible feedback when brain activity patterns representing speech production or imagery are detected and silence otherwise. This way, no additional speech onset detection is necessary. Our approach is implemented as a node-based framework that enables the organization of individual processing steps as self-contained units that can be assigned to a new process for parallel computing. Figure 2 shows a schematic overview of our proposed closed-loop synthesis approach for real-time speech decoding, summarized as follows:

a. Neural signal acquisition: Invasive brain signals related to speech processes are acquired directly from the brain through implanted sEEG electrodes. An input node of the system is connected to the amplifier through LabStreamingLayer[20] which serves as the middleware between the recording setup and the decoding software.

b. Processing of neural signals: The multichannel neural signals are processed to extract high-gamma power, which correlates with ensemble spiking[21] and contains highly localized information about speech processes[22,23]. While previous speech decoding studies[11,24] have generally considered a large temporal context, both before and after speech production[25], real-time processing cannot consider neural data of temporal context after speech production to avoid delayed feedback that would adversely impact the natural speech production process[26].

c. Decoding models: The decoding models assign vectors of neural activity to classes of discretized audio spectrograms. For this purpose, the audio spectrogram is mel-scaled[27] using 40 triangular filter banks. In each spectral bin, the signal energy is discretized into nine energy levels based on the sigmoid function[28]. This quantization enables a high resolution in the spectral range that contains the participant's voice but also provides sufficient information to reliably represent silence. Each of these nine energy levels represents one of the target classes in the classification approach. Regularized LDA-classifiers then predict the energy level for each of these spectral bins. Thus, the system consists of 40 LDA-classifiers that are trained on the open-loop data from the first experimental run. This simple classification scheme was selected to ensure the generated output remains within typical boundaries of speech production and is more robust against noise and outliers in the neural data. To keep the dimensionality for this

**Fig. 1 Overview of experimental design.** The experiment begins with an open-loop run in which the participant reads a series of 100 words aloud while the speech and brain activity are synchronously recorded. In the two subsequent closed-loop runs, the participant performs the same task while whispering and imagining speech, respectively. For the closed-loop runs, real-time audible feedback of the neurally-decoded and synthesized speech is provided via our system.

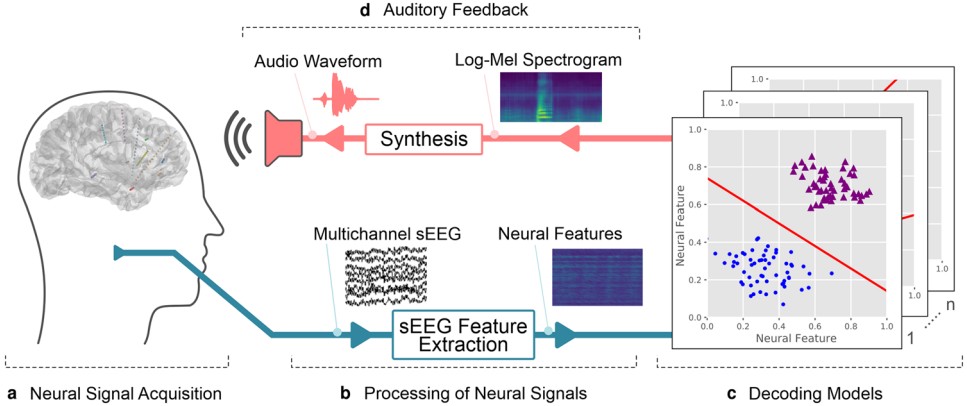

**Fig. 2 Schematic overview of our proposed real-time synthesis approach. a** Invasive brain signals are acquired through implanted sEEG electrodes. **b** Multichannel signals are processed to extract the high-gamma power. **c** Linear decoding models are used to estimate a spectral representation (**d**) which is synthesized into an audible speech waveform using the Griffin–Lim algorithm and presented to the patient as real-time auditory feedback.

classification problem manageable, 150 features were selected based on the correlations with the speech energy.

d. Auditory feedback: Generation of an audible speech waveform is implemented via the Griffin–Lim algorithm[29] which reconstructs the phase spectrogram iteratively. We constrained the approximation to 8 iterations based on the results of a prior study[30]. The reconstructed speech waveform is presented to the participant as auditory feedback over the integrated loudspeakers of a dedicated research laptop.

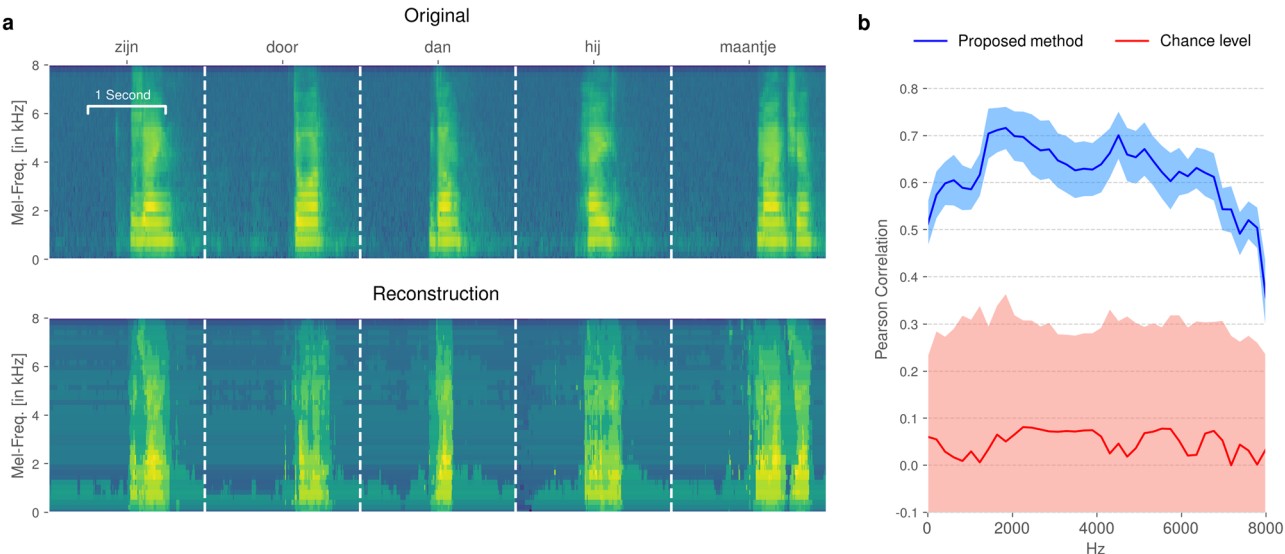

**Fig. 3 Decoding performance of the proposed method on the open-loop data from the audible speech experiment.** The spectrogram was reconstructed using ten-fold cross-validation. **a** Visual comparison of original and reconstructed spectrograms. **b** Correlation coefficients across all spectral bins for our approach (blue, $n_1 = 10$) compared to a randomized baseline (red, $n_2 = 100$) generated by breaking the temporal alignment between the brain signals and speech recordings. Shaded areas represent standard deviation.

## Results

**Decoding performance in the audible open-loop experiment.** We first quantified the performance of the general decoding concept on the data from the open-loop data run. The evaluation was performed retrospectively as these data were collected for model training for the closed-loop runs. We used ten-fold cross-validation to reconstruct the complete speech spectrogram of the entire open-loop run, including data from all trials and intertrial intervals. Comparisons between speech spectrograms were quantified by average Pearson correlation coefficient across all frequency bins. We evaluated our output using the Pearson correlation coefficient instead of measures of automatic intelligibility, like the STOI[31] or ESTOI[32], as certain trials might not satisfy the assumption of 400 ms of consecutive speech. Currently, reconstructed speech is not intelligible. Figure 3a shows a comparison between the original and reconstructed spectrograms. Examples have been selected based on the top five highest Pearson correlation scores (left 0.89, others: 0.82) and are presented for visual inspection. Overall, we achieve an average correlation of $0.62 \pm 0.15$ across all spectral bins (blue line, Fig. 3b). As a baseline, we established a chance level (red line, Fig. 3b) by artificially breaking the temporal alignment between neural activity and acoustic speech and then retraining the classifiers with this broken alignment: We split the acoustic data at a random time point into two partitions and temporally swapped the resulting partitions. We then retrained the entire decoding pipeline using this new alignment, in which the neural data should not contain any information about the misaligned audio. This procedure provides a good estimation of chance-level reconstruction results. We repeated this procedure 100 times and compared the distribution of correlation coefficients with those from the real data using Mann–Whitney $U$ tests. The proposed method significantly outperforms the baseline in every frequency bin (Mann–Whitney $U$ test, $P < 0.001$, $n_1 = 10$ and $n_2 = 100$, Bonferroni corrected). While our decoding approach achieves mean correlation scores above 0.6 for the majority of frequencies involved in human speech, the mean chance level remains consistent around 0.1.

**Closed-loop synthesis of whispered speech.** For the closed-loop runs, the whispered speech condition provides an initial feasibility check as no audible vocalization is present, ensuring that the decoding approach is not impacted by acoustic speech contamination[18].

During the experimental run, the closed-loop decoder synthesized the measured neural activity into an acoustic waveform that is presented to the patient in real-time and recorded for offline evaluation. Figure 4a shows decoded audio waveforms for five selected trials. These examples indicate that the closed-loop decoder was capable of reconstructing audible audio with onset/offset timings that are reliably gated by neural correlates of whispered speech processes and are characteristic of natural speech.

In contrast to the open-loop run, the patient only produced a barely perceptible vocalization in the whispered run, which was below the sensitivity of the microphone. Therefore no reference is available for the comparison of synthesized and actual speech. However, 73 random target words appeared in both the open-loop and whispered runs and therefore the actual articulations are available for comparison, although not precisely time-aligned between runs. In order to quantify the decoding performance, we examined the Pearson correlation coefficient between decoding results from whispered trials and time-warped reference articulations: for each word, we optimized the temporal alignment by warping the mel-scaled spectral representation of the reference articulation to the decoded spectral coefficients using a dynamic-time warping (DTW) algorithm[33]. Based on these alignments, Pearson correlation coefficients were computed. To establish a chance level, we applied our decoding approach to randomly selected 2-s segments of neural data acquired during a separate session of nonspeech tasks (grasping and eye fixation) performed by the participant. This procedure was repeated 1000 times to create a reliable distribution. Our approach achieved median DTW correlations of $0.64 \pm 0.15$ and $0.17 \pm 0.23$ for the chance level (Fig. 4c). Mean correlation coefficients across all spectral bins are significantly higher than the chance level ($P < 0.001$, Mann–Whitney $U$ test, $n_1 = 73$ and $n_2 = 1000$). These correlation

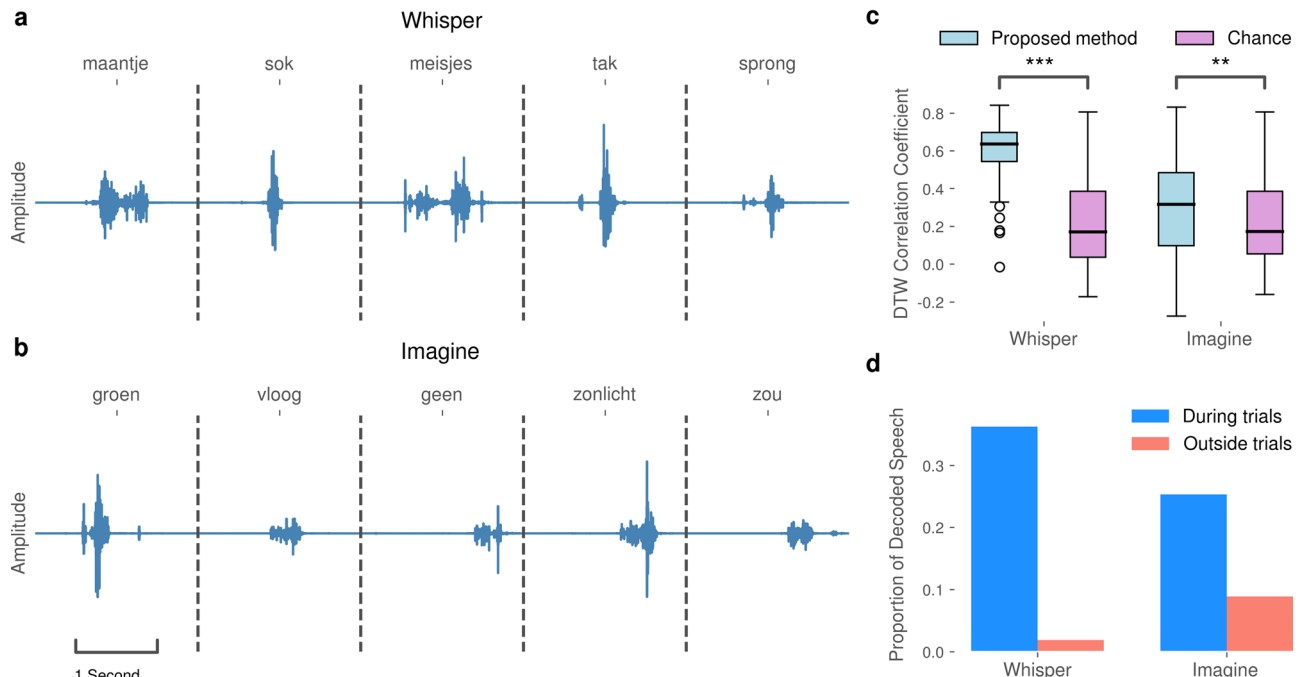

**Fig. 4 Decoding results of the proposed method in the closed-loop experimental runs. a** Selected examples of synthesized audio waveforms produced during whispered speech trials. **b** Selected examples of synthesized audio waveforms produced during imagined speech trials. In both runs, the speech was reliably produced when the participant was prompted to whisper or imagine to speak, respectively. **c** Pearson correlation coefficients between time-warped reference speech trials and closed-loop whispered trials ($n_1 = 73$) and closed-loop imagined speech trials ($n_1 = 75$), respectively. Chance level ($n_2 = 1000$) is based on randomly selected data from non-speech tasks performed by the participant. Statistical significance, indicated by asterisks (***$P < 0.001$; **$P < 0.01$), was computed using Mann–Whitney $U$ tests. Black horizontal lines correspond to median DTW correlations scores. Boxes define boundaries between the first and the third quartile. Error bars present the range of data within 1.5 times the interquartile range, and points beyond the range of the error bars show outliers. **d** The proportion of decoded and synthesized speech during whispered and imagined trials, respectively, versus non-speech intertrial intervals.

coefficients are not directly comparable to those shown in Fig. 3b, due to the optimal temporal alignment imposed by the DTW. In an additional analysis, we investigated the proportion of decoded speech during trials and speechless intertrial intervals (Fig. 4d). We utilized a voice activity detection (VAD) algorithm[34] that automatically annotates speech segments based on energy levels in an acoustic waveform. Our approach reliably decoded and synthesized speech during the majority of whisper trials and not during the speechless intertrial intervals. Only a small proportion of speech was decoded and synthesized during speechless intertrial intervals. Despite the fact that the closed-loop decoder is not trained on whispered speech processes, the generated acoustic waveforms demonstrate successful decoding, which is in accordance with prior findings regarding the transferability of decoding models towards mimed speech[8]. However, the synthesis performance is worse than the performance on audible speech processes—likely due to the absence of actual phonation, which was also hypothesized in the previous work[8].

Based on our causal design, the decoding results do not rely on perceived auditory feedback of vocalizations and are based on neural processes underlying speech production.

**Closed-loop synthesis of imagined speech.** For the imagined speech run, we performed the same evaluation steps as for the whispered speech data. Previously, it had been unclear whether models trained on neural activity during audible speech can be transferred to imagined speech processes. Imagined speech, in contrast to both whispered and audible speech, does not involve any articulator movements of the vocal tract. Figure 4b depicts five promising example acoustic waveforms resulting from the real-time synthesis of imagined speech processes. The decoded

speech gating is comparable to the whispered speech condition, which indicates the transferability of the actual speech model to imagined speech. As with the whispered speech condition, Fig. 4c reports the DTW correlation coefficients between decoded trials and their reference vocalization. For this case, 75 random words overlapped with the open-loop run. We achieve mean correlation coefficients of $0.32 \pm 0.26$ for our proposed method, which is significantly higher than the chance level of $0.17 \pm 0.22$ ($P < 0.01$, Mann–Whitney $U$ test, $n_1 = 75$ and $n_2 = 1000$). However, compared to whispered speech, the decoding performance is significantly lower ($P < 0.001$, Mann–Whitney $U$ test, $n_1 = 75$ and $n_2 = 73$). The majority of speech segments as identified by VAD still occurred during the imagined trials (Fig. 4d); however, there were a larger number of speech segments during the nonspeech intertrial intervals compared to the whispered condition. This is likely due, in part, to the inherent difficulty of knowing how the participant internally performed imagined speaking, in conjunction with the differences between actual and imagined speech processes. These results highlight that the models trained on audible speech can be used to synthesize imagined speech processes in real-time. The provided feedback will provide important queues for allowing the user to learn and adapt to the system, and is ultimately necessary for a practical prosthesis.

**Anatomical and temporal contributions.** The design of our real-time decoder only utilizes neural activity prior to speech production for the decoding, as a delay would be introduced otherwise. The participant was implanted with a total of 11 electrode shafts containing a total of 119 electrodes. Electrodes were predominantly implanted in the left frontal and lateral areas, except for the purple and red electrode shafts in the temporal region

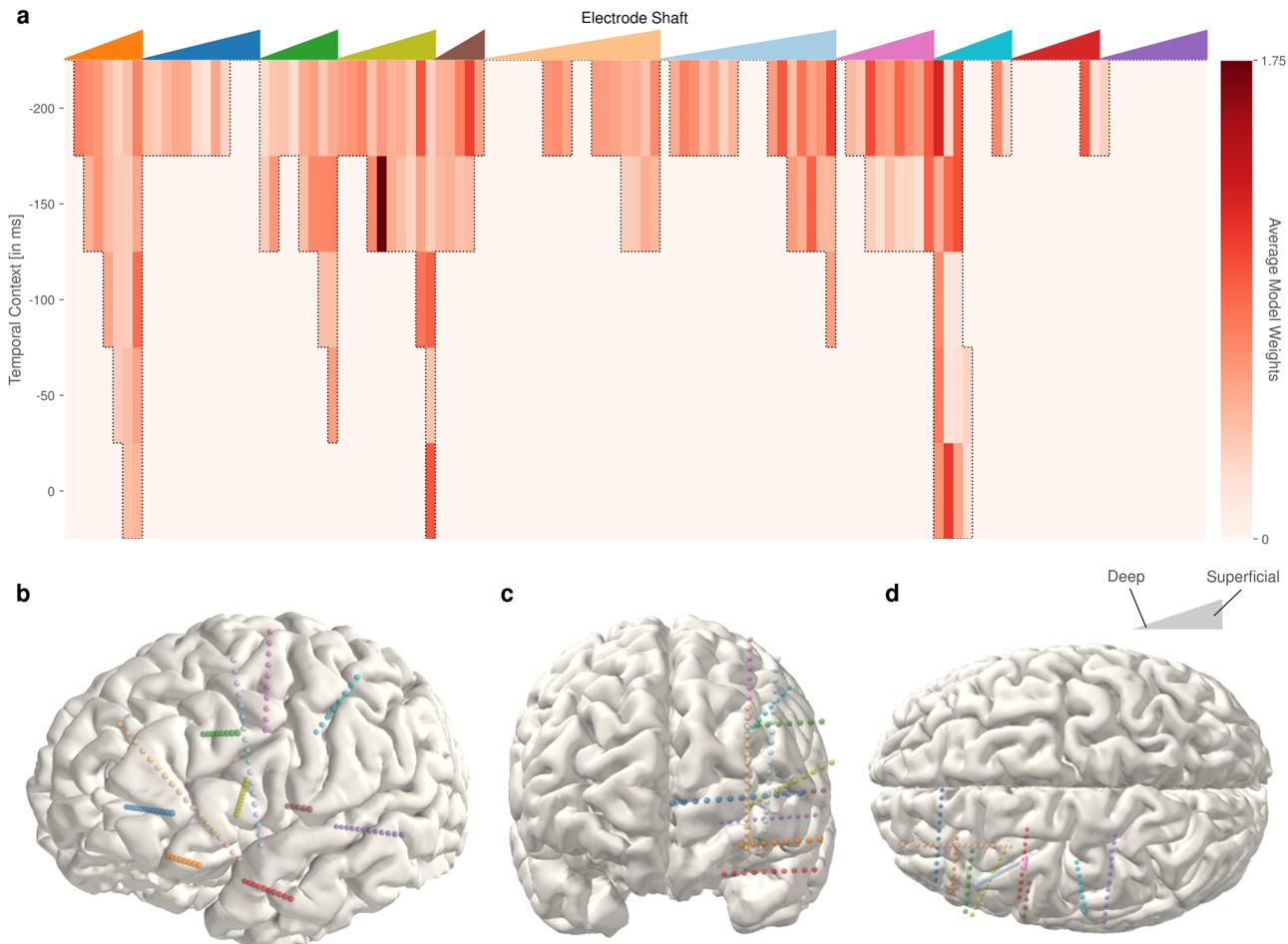

**Fig. 5 Anatomical and temporal contributions. a** Spatiotemporal decoder activations averaged across 9 classes and 40 frequency bins. The colored triangles at the top of the panel correspond to the colored electrode shafts in (**b–d**). The left edge of each triangle represents the deepest contact of a respective colored electrode shaft and the most superficial contact the same shaft is represented at the right edge, with the intermediate contacts ordered longitudinally in between. The activations (i.e., transformed average model weights) for the decoding models at each electrode and temporal lag are indicated by the vertical bars below the corresponding colored triangle. Darker red indicates higher absolute activations. The activations indicate that inferior frontal and middle frontal cortices are predominately employed in decoding. **b–d** Different views of electrode locations for the participant: **b** left lateral, **c** frontal, **d** superior.

(Fig. 5b–d). Features from the 119 electrode contacts across the five different temporal contexts (−200 to 0 ms offset) were selected based on the correlations of high-gamma power with the speech energy. For the selected features, we visualize the average absolute activation across all LDA models, using the methods described by Haufe et al.[35] to analyze the contribution of different functional areas and temporal context (Fig. 5a). Functional areas were identified using img_pipe[36] and FREESURFER[37].

The decoding models predominately rely on high-gamma activity from 150 to 200 ms prior to the current time point, which is in accordance with prior research in the role of the frontal cortex in speech production[25]. In particular, areas in the inferior frontal gyrus triangularis (olive green) and opercularis (brown) showed high activations. These relevant features also include activations representing the current time point (i.e., 0 ms), such as the superficial contacts in the orbital gyrus (orange). Electrodes in the anterior cingular sulcus and gyrus and frontomarginal cortex (blue) contributed comparatively small activations. Superior frontal and middle frontal gyrus (green, outer electrodes of beige, light blue, pink) were used for decoding from 200 to 50 ms prior to the current time point. Electrode contacts in the anterior insular (deeper contacts of beige) were not selected for the decoding. In addition to the aforementioned electrodes likely involved in speech planning[38,39], electrode contacts in the central sulcus and postcentral gyrus (turquoise), showed high activations from 200 ms prior up to the current time point, likely reflecting activity associated with articulator control. Limited activations were also observed in the superior temporal sulcus (red) and hippocampal contact (deepest contact of purple).

These activation patterns trained on the open-loop run with audible speech also consistently decode speech from the whispered and imagined speaking modes, which point to a role in speech planning and execution as opposed to the perception of the participant's own voice. Differences in neural activation patterns have been reported for different acoustic and proprioceptive feedback conditions[40,41], but our decoding approach still identifies patterns that are similar enough across the three speaking modes to generate consistently gated synthesized output. This further implies that speech production and imagined speech production share a common neural substrate to some extent, which has also been proposed previously[42].

## Discussion
Here, we demonstrated that intracranial recordings can be used to synthesize imagined speech processes in real-time as continuous acoustic feedback. Through the successful use of sEEG electrodes,

we establish feasibility for long-term implantation, as these electrodes are comparable to those used for Deep Brain Stimulation (DBS). DBS procedures are commonplace for a variety of neurological conditions (e.g., Parkinson's disease, essential tremor, dystonia) and the electrodes routinely remain implanted and effective for decades[43]. Furthermore, the implantation of such stereotactic depth electrodes has a low-risk profile, as a craniotomy is not required[44–46].

Our decoding models rely predominantly on high-gamma features from the frontal cortex and motor areas, which have been implicated in existing models of speech production[38,47,48]. This also offers an explanation as to the transferability of the decoding models to whispered and imagined speech. While these three processes are clearly different in terms of acoustic and proprioceptive feedback[41], our results indicate that they share enough neural activity to enable model transfer. The selected areas can also function as a blueprint for future implants in patient cohorts.

Our approach still requires audible speech for training the decoding models. Several possibilities exist for moving towards clinical practice where pre-recorded speech from the patient may not be available. First, electrodes could be implanted in the early stages of neurodegenerative disease (e.g., ALS) such that training data can be acquired while the patient still has the capacity for speech. Alternately, surrogate data could be created by asking the patient to imagine speaking along with previously recorded audio, from the patient or potentially other speakers.

In this study, we have intentionally focused on a simplistic decoding approach specifically to achieve the proof-of-concept of real-time acoustic feedback from neural activity and thereby tackle closed-loop synthesis of imagined speech. While the synthesized output was able to effectively generate and gate audible speech in real-time with utterance timings comparable to the vocalized speech reference, the output is not yet intelligible and quality is expectedly inferior to prior results based on offline decoding of audible and whispered speech[6–8]. For a demonstration of the real-time synthesis, see Supplementary Movie 1. However, the fact that our models were trained on actual speech and were successful in triggering the synthesized output during imagined trials in real-time, provides a strong indication that we have tapped into common underlying processes that can be further exploited by utilizing more sophisticated decoders, such as deep neural networks (DNN)[6,8,49]. While certain trained DNNs are capable of real-time execution, it is envisioned that the user and system will need to co-adapt to optimize performance. Thus, there is utility in continuing to explore simplified decoding models that can provide some degree of real-time adaptation as we continue to understand the nature and dynamics of the underlying speech processes and human factors of the system. The immediate feedback provided by our system will allow patients to learn to operate the prosthesis and improve synthesis quality gradually. In addition, we only investigated the production of individual, prompted words. In the future, spontaneous speech processes need to be investigated to move toward natural conversation.

Our approach demonstrates real-time speech synthesis from imagined neural activity and provides a framework for further testing and development. These initial results of decoding imagined speech in real-time represent an important step forward toward a practical speech neuroprosthesis for those who are unable to speak. By synthesizing imagined speech processes, we are able to actively engage the participant in the imagined speech experiment, effectively allowing the participant to monitor the speech production process and adjust accordingly[40]. This process will be critical as the user learns to use and control the neuroprosthetic, and eventually for co-adaptation between the user and system[50]. These initial results provide evidence that neural activity during speech imagery can be used in the development of future speech neuroprosthesis.

## Methods

**Participant**. In the medical treatment of a severe epilepsy patient (female, 20 years old) 11 sEEG electrode shafts, with 8 to 18 contacts, were implanted into the left hemisphere. Electrodes were implanted to determine the epileptic foci and map cortical function to identify critical areas for which resections might result in long-term functional deficits. During this monitoring process, the patient agreed to participate in scientific experiments. The patient gave written informed consent and participation in the experiment was on a voluntary basis. Participation could be terminated at any time by the patient without giving any reason. The experiment design was approved by the IRB of Maastricht University and Epilepsy Center Kempenhaeghe and was conducted in a clinical environment under the supervision of experienced healthcare staff. The participant was a native speaker of Dutch.

**Electrode locations**. Electrode placement was purely determined based on clinical needs and in no way influenced by the research. Electrode locations were determined by co-registering a pre-operative T1-weighted MRI with a postoperative CT scan of the participant. Co-registration and anatomical labeling were performed using FREESURFER[37] and img_pipe[36], where anatomical labels of the contact locations were assigned after cortical parcellation according to the Destrieux atlas[51].

**Data recording**. The implanted platinum-iridium sEEG electrodes (Microdeep intracerebral electrodes; Dixi Medical, Beçanson, France) were 0.8 mm in diameter and containing 8–18 contacts. Electrode contacts were 2 mm in length and had 1.5 mm inter-contact distance.

Activity from stereotactic EEG electrodes was recorded using a Micromed SD LTM amplifier (Micromed S.p.A., Treviso, Italy) with 128 channels, referenced to a common white matter electrode. Data were digitized at 2048 Hz. Audio data were recorded using the integrated microphone of the recording notebook (HP Probook) at 48 kHz. Audio data and neural signals were synchronized using LabStreamingLayer[20].

**Task**. The experimental task consisted of three sequential experimental runs. For the first run, we acquired open-loop neural signals and audibly vocalized speech in parallel as training data to optimize the parameters of the closed-loop synthesizer. For each trial in the run, the patient read individual Dutch words aloud as they were presented sequentially on a monitor. Each word was shown for 2 s with an intertrial interval of 1 s. Words were from the Dutch language and were drawn randomly from a set comprised of phonetically balanced words from the IFA dataset[19] and the numbers 1–10. In each experimental run, a total of 100 words was prompted resulting in ~5 min of data per experimental run.

Subsequently, two closed-loop runs were performed. The patient repeated the open-loop task using whispered and imagined speech, in respective runs, while the closed-loop system continuously decoded the neural signals and provided acoustic feedback in real-time. All participant–system interactions were logged for further offline analysis and evaluation of the experimental results.

**Neural signal processing**. In line with prior studies[6,28,30], we focused on neural activity in the high-gamma (70–170 Hz) band, which is known to contain highly localized information relevant to speech[22,23] and language[52] processes. High-gamma activity also provides detailed information about speech perception processes in ECoG[53,54] and stereotactic EEG[49]. For the extraction of the high-gamma band and for the attenuation of the first and second harmonic of the 50-Hz line noise, we used an IIR bandpass filter and two elliptic IIR notch filters, respectively, each having a filter order of 8. The resulting signals were segmented into 50 ms windows with a 10 ms frameshift to capture the complex dynamics of speech processes. For each of these windows, we calculated signal power and apply a natural logarithm transform to make the distribution more Gaussian prior to decoding. In addition, each window was augmented with context information up to −200 ms in the past to integrate temporal changes in the neural dynamics. Since we are targeting a real-time system response, context stacking of future frames[55] is not utilized to avoid additional latency.

**Audio signal processing**. Time-aligned recordings of spoken speech captured by an integrated microphone in the research laptop were transformed to logarithmic mel-scaled spectral features using the following procedure: We first downsampled the audio data to 16 kHz and segmented the signal into overlapping windows with a window length of 16 ms and a frameshift of 10 ms. Subsequently, we used triangular filter banks to compress the frequency range based on the perception of pitches by means of the mel scale. This preprocessing procedure aligned sEEG features and spectral acoustic features while preserving speech-relevant information, and results for each window in 40 logarithmic mel-scaled spectral coefficients.

Based upon prior research[28], we discretized the feature space of the spectral coefficients to avoid the unbounded characteristics of regression tasks, which can

result in unintended large amplitude spikes in the audio feedback caused by neural fluctuations. As the experiment design contained only single words, the distribution of windows representing speech and silence frames was highly imbalanced in favor of silence and therefore we incorporated a quantization based on the curve value of the logistic function to find suitable interval borders:

$$f(x) = \frac{|\min_{spec}| + \max_{spec}}{1 + e^{-k \cdot x}} - |\min_{spec}| \tag{1}$$

Here, $\min_{spec}$ and $\max_{spec}$ represent the minimum and maximum power value for each spectral bin and the hyperparameter $k$ defines the growth rate. Using a uniform spacing for the independent variable $x$ results in interval borders which are prioritizing both coefficients in the low-frequency range, such as silence, as well as in higher, speech-related, frequencies and are unaffected by their imbalanced distribution.

For this study, we used a fixed number of nine intervals for each spectral bin and a constant growth rate of $k = 0.5$ based on prior experiments with offline data[28].

**Classification approach**. The mapping from neural data onto spectral coefficients is relying on information encoded in the high-gamma band. Model training is designed to identify these correlates with respect to underlying speech processes. In previous studies, we have used regression approaches as our decoding models[6,30] to enable a conversion directly projecting onto the continuous space of spectral coefficients. However, we observed substantial amplitude variation in the decoded acoustic waveform resulting from the unbounded behavior of regressions in conjunction with large fluctuations in the neural signals. Here, we focus on modeling strategies that decode waveforms with a constant volume to avoid such unnatural spikes in loudness. Therefore, we expressed the regression task as a classification problem: rather than establishing a direct mapping, we discretized the continuous feature space of each spectral bin into a fixed set of nine energy levels. Individual classifiers were then used to predict the energy levels in each individual spectral bin. To recover the speech spectrogram, we applied a dequantization step that replaces all indices with proper spectral coefficients represented by each interval. For the classifiers, we utilized a regularized linear discriminant analysis (LDA) implemented in scikit-learn[56]. We utilized singular value decomposition as a solver as it is known to perform well with high-feature dimensionality. The classifiers were trained on the 150 features with the highest correlation with the audio energy on the training data. We chose 150 features as a good compromise between the amount of included information and decoding speed based on prior experience.

**Voice activity detection**. We reimplemented the energy-based voice activity detection from the Kaldi speech recognition toolkit[34] to distinguish between speech and nonspeech segments in the decoded audio (Fig. 4d). This approach relies on the signal energy of mel-frequency cepstral coefficients and classifies individual frames based on a threshold. The method uses four customizable hyperparameters: a constant term $t$ for the energy threshold, a scaling factor $s$ for the energy level, the number of augmented context frames, and a threshold of the required proportion $P$ of speech frames inside a window. To calculate the threshold $T$, we take the product of the mean energy in the frequency cepstrum and the scaling factor and add the specified constant term:

$$T = t + s \cdot \frac{1}{N} \sum_{i=0}^{N} E_{0,i} \tag{2}$$

Here, $E_0$ refers to the first coefficient in the frequency cepstrum and $N$ corresponds to the number of frames. In the next step, we assign to each frame a number of temporal context frames leading and lagging the current point to form a window, where it is possible to have unequal numbers of lagging or leading frames. The identification of speech frames is based on the ratio of how many frames in each window are above the threshold. If this ratio is greater than the required proportion $P$ it gets marked as speech.

We have identified suitable hyperparameters by only considering the audio of the training data, which we then applied to the decoded audio for evaluation. We used an energy threshold of 4 with a mean scaling of 0.5. In addition, we limited the maximum amount of context frames to five on each side and required a ratio of at least 60% of frames inside a window marked as a speech before being identified as speech.

**Activation maps**. To determine which cortical areas contributed to the decision of individual classes from the linear discriminant analyses, we employed a method proposed by Haufe et al.[35] that transforms weight vectors of linear decoding models into a form for neurophysiological interpretation. Here, we will briefly summarize the steps taken to extract neural activation maps from our multivariate classifiers. Based on the internal weights $W$ of each decoding model $i$ and the measured neural signals $x(n)$ from the audible experiment run, we infer latent factors $\hat{s}(n)$ by carrying out a simple multiplication:

$$\hat{s}(n)_i = W_i^T \cdot x(n), \tag{3}$$

where $n$ specifies the number of samples. Here, the latent factors exhibit certain properties with respect to the supervised classification of target variables. The

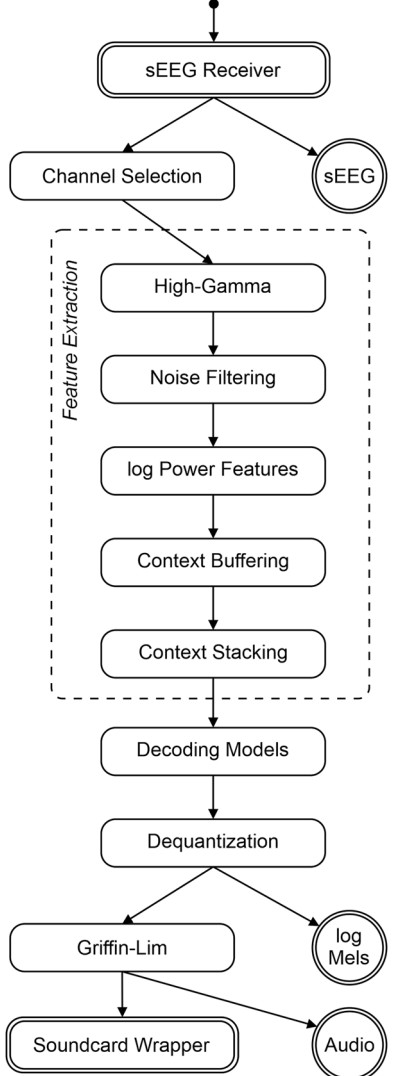

**Fig. 6 Schematic overview of the proposed closed-loop decoder.** Each node corresponds to one self-contained task, which is connected in an acyclic network. Rectangular nodes specify actual computations in the generation of the acoustic waveform, while circular nodes represent output nodes writing incoming data on disc for offline evaluation. Double-lined nodes indicate that its (and subsequent) calculations are performed asynchronously in different processes. The extraction of neural activity is composed of multiple nodes.

method by Haufe et al. shows that activation patterns regarding all target variables can be extracted in the following way:

$$A_i = \Sigma_x \cdot W_i \cdot \Sigma_{\hat{s}_i}^{-1}, \tag{4}$$

where $\Sigma_x$ and $\Sigma_{\hat{s}}^{-1}$ correspond to the covariance matrix of the measured neural signals and the inverse covariance matrix of the latent factors, respectively. In order to draw our conclusions regarding the anatomical contributions, we took the mean absolute activations across all decoding models and interval classes.

**Closed-loop architecture**. The closed-loop decoder is built upon a node-based system, in which each node performs the calculations of a self-contained task and multiple nodes are connected in a directed acyclic network which specifies the processing chain. Our entire closed-loop system is implemented in custom Python code. Each node can have an arbitrary number of other nodes feeding input data, and can itself feed its output to an arbitrary number of other nodes. Data are passed as two-dimensional matrices, with the first dimension representing time. Nodes do not need to operate at the same rate, a node can be implemented to output fewer (e.g., in the case of windowing) or more time steps of data than it accepts as input. In addition, to optimally take advantage of modern multicore hardware and to reduce processing delays, nodes can be implemented to output data

**Table 1 Mean processing costs for each node in the decoding pipeline.**

| Processing step | Processing costs [$\frac{ms}{per\ 10\ ms}$] |
|---|---|
| Channel selection | 0.137 |
| High-Gamma | 0.773 |
| Noise filtering | 0.766 |
| Log power features | 0.597 |
| Context buffering | 0.406 |
| Context stacking | 0.379 |
| Decoding models | 2.008 |
| Dequantization | 0.612 |
| Griffin–Lim | 1.652 |
| **Total** | **7.331** |

asynchronously, with calculations being performed in a process different from the process in which input is received.

Figure 6 shows individual nodes and their connections assembling the closed-loop decoding pipeline. While the chain of rectangular nodes constitutes the pathway for the conversion from incoming neural signals to an acoustic waveform, circular nodes represent output nodes for writing intermediate data to disc for offline evaluation. In addition, double-lined nodes indicate the starting point for asynchronous computations of subsequent nodes in different processes. The input receiver represents a socket that listens for incoming packets from the amplifier via LabStreamingLayer. The amplifier is configured to send 32 packets of sEEG data per second. In a first step, any questionable channels (e.g., floating, contaminated, etc.) are removed by excluding channel indices that have been manually identified prior to model training (no channels were identified for exclusion in the current study). All filters for extracting log power from the high-gamma band are designed to work on streamed data by preserving the filter state. The decoding models predict spectral interval classes independently for each frame and the dequantization uncovers the spectral coefficients for 10 ms per frame. The Griffin–Lim algorithm is implemented using two ring buffers in order to run properly for streamed data. The first ring buffer, which is responsible for the input, stores a limited context of 65 ms of previous frames, while the second ring buffer carries the output signal, from where the corresponding segment is extracted and passed to the soundcard wrapper for audible presentation.

In Table 1, we present mean processing times measured on the research laptop for each individual node in the decoding pipeline to ensure real-time capabilities. All nodes present low processing costs and do not constitute a bottleneck in the system. In total, for 10 ms of neural signals, 7.3 ms are needed for the conversion into an acoustic signal. For the decoding pipeline, we only used one process to avoid cost-expensive inter-process communication. Output nodes for intermediate results, on the other hand, operate in dedicated processes. In addition, we set up a separate process to listen for markers sent during each experiment run to align the timings of each trial.

**Performance metrics.** Comparisons between time-aligned pairs of decoded and original speech spectrograms are determined using the Pearson correlation coefficient. For each spectral bin, we calculate the correlation between both time series of logarithmic mel-scaled spectral coefficients and compose a final score based on their mean. In the evaluation of whispered and imagined speech, direct comparison based on the correlation is not possible as no time-aligned reference is present. We, therefore, warp the reference data first to obtain an optimized temporal alignment using dynamic-time warping (DTW). By computing the shortest path based on the Euclidean distance and rearranging frames of spectral coefficients in the reference data, we achieve a warped representation with minimal distance. With this warped representation, the calculation of correlations is possible again. It should be noted that the time-aligned correlations and time-warped correlations cannot be directly compared, as the warping artificially maximizes correlations.

**Statistics and reproducibility.** The differences between our proposed method and a chance level or the differences between two speaking modes were evaluated by two-sided Mann–Whitney U test statistics using the scipy stats package (version 1.5.3). We provide the experiment data and the code which can be used to reproduce the results and figures.

**Reporting summary.** Further information on research design is available in the Nature Research Reporting Summary linked to this article.

## Data availability
All physiological and empirical data can be obtained from https://osf.io/cng9b/. Note that we anonymized the patient's voice. Data to reproduce Fig. 4c, d can be found in the same repository.

## Code availability
Custom code for this research was written in Python 3.6 programming language. Code for the closed-loop synthesis approach, as well as all analysis scripts can be obtained from https://github.com/cognitive-systems-lab/closed-loop-seeg-speech-synthesis. This repository also includes the code for rendering the plots shown in Figs. 3–5. We conducted the closed-loop experiment on a laptop running Windows 7 and performed the analysis on an Ubuntu 18.04 computer.

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

## Acknowledgements
C.H. acknowledges funding by the Dutch Research Council (NWO) through the research project 'Decoding Speech In SEEG (DESIS)' with project number VI.Veni.194.021. T.S., D.J.K., and M.A. acknowledge funding by BMBF (01GQ1602) and NSF (1902395, 2011595) as part of the NSF/NIH/BMBF Collaborative Research in Computational Neuroscience Program.

## Author contributions
C.H., M.A., P.K., A.J.C., and L.W. conceived the experiment. C.H. and M.O. recorded the data. M.A. and C.H. analyzed the data. M.A., L.D., D.I., and G.I. implemented the closed-loop synthesis approach. T.S. and D.J.K. supervised the development of the closed-loop synthesis approach. S.G. and J.S. localized the electrodes. M.A. and C.H. wrote the manuscript. All authors reviewed the manuscript.

## Competing interests
The authors declare no competing interests.
