## [Peer Review File · Communications Biology]

Reviewers' comments:

Reviewer #1 (Remarks to the Author):

This manuscript is about real-time synthesis of imagined speech processes from invasive neural recordings. One participant was included.

In general the manuscript is well written and good to understand. I could not identify any conceptual mistakes, however, I have some questions/remarks:

- not clear for me is how the onset of the speech is detected - especially in the imagined speech.
- the classification approach is not fully clear. What were the classlabels to train the LDA? This section needs some more sentences so that a reader can better understand what you exactly did.
- Figure 3B: statistics for each frequency bin: did you account for multiple testing?
- did you always give the words to be spoken in advance? did you try to let the participant imagine words that you did not know in advance?
- The results are impressive, however, at least for a non-dutch listener hard to assess. You could let dutch people listen to the synthesized words and let them rate what they hear. This would give another picture to a reader, how good the quality of the speech recognition and synthesis is.

Reviewer #2 (Remarks to the Author):

Brief summary

The authors recorded brain activity from stereotactic EEG (sEEG) electrodes implanted into the brain of a young female patient who read aloud words under three different conditions: 1. overt production, 2. whispered production, and 3. imagined production. Condition 1 includes articulatory-motor information and produces an acoustic signal; condition 2 produces hardly any acoustic signal but includes articulatory-motor information; condition 3 is just the imagined speech movement, no articulatory-motor nor any acoustic signal.

Neural activity in the high gamma band is extracted, decoded and used to transform the neural activity into spectral output (auditory speech waveform), which is then fed back to the speaker. The decoding model is trained on the overtly produced speech; the whispered speech is then used as a test case without acoustic signal (but including motor signals). The imagined speech condition is the most relevant condition in this experiment.

Even though the spectral output signal (see video and audio file provided by the authors) is not comprehensible yet, in my opinion this article provides innovative and very important data. For patients who can no longer communicate due to their illness, this method may present a possible solution. This article provides the proof of concept.

Critical evaluation

The authors claim on the basis of their findings "that speech production and imagined speech production share a common neural substrate" (lines 168-169). However, in the neurolinguistic literature on speech production and speech monitoring (see e.g. Christoffels et al., 2007, HBM; also Christoffels et al., 2011, PLoS One) overt and covert (i.e. imagined) speech production lead to huge differences in brain activity, not only in motor-related areas. Interestingly, the current paper claims that the neural signal for speech planning (high gamma band activity) recorded from the electrodes

in the three different conditions was comparable or similar enough for the decoder – trained on overt speech – to extract the necessary information in the whispered and imagined speech conditions. I believe it is worth mentioning this discrepancy and explain why this discrepancy exists. Regarding their methodology, the authors need to make clear what the difference between closed- and open-loop experiments/conditions is. In the overt speech production condition, the patient read aloud the words on the screen and heard herself (via the internal-cognitive as well as external – acoustic and bone transmission – monitoring loop). In the whispered condition, this is comparable, only that the acoustic signal is much weaker and hence also the bone transmission. However, it is not clear to me why the overt condition is labeled as “open loop” and the whispered condition as “closed loop”. Finally, in the imagined naming condition, the patient can only monitor herself via the internal monitoring loop (since no motor movements are carried out and no acoustic output is produced). Therefore, the imagined speech condition may be a closed loop in the sense that there is no external monitoring for the patient. The difference between open- and closed-loop experiments should be made explicit to avoid confusion.

It is well established, by the way, that when speakers monitor their output even when they do not produce any sounds. This is called internal monitoring (see e.g. Levelt et al., 1999, BBS). A potential problem with the imagined speech condition is that the authors cannot be sure that the patient cooperated and how well she imagined producing the words covertly.

As mentioned above, the “reconstructed speech is not intelligible” (line 88). Given this fact, the authors need to state the purpose of the feedback, i.e. why was the reconstructed speech played back to the patient if it was incomprehensible anyway? I could imagine that once the signal can be improved feedback would fulfil a function, however, under the current conditions, the purpose is not clear to me.

Taken together, this is a well written report of a proof of concept that acoustic output can be reconstructed on the basis of neural signals alone. This may be an important step in the direction of developing a speech neuro-prosthesis based on imagined speech for patients who are no longer able to communicate verbally with their environment. The report is very innovative and potentially groundbreaking.

Dear Reviewers,

We thank you for your thorough review and very helpful comments. We were particularly happy to read that our work is “well written” and that the reviewer “could not identify any conceptual mistakes” (reviewer 1). Furthermore, reviewer 2 described our work as “very innovative and potentially groundbreaking”.

We feel that by addressing your comments, the manuscript has improved substantially. In the following, we will describe how we addressed each comment individually.

Reviewer #1

- not clear for me is how the onset of the speech is detected - especially in the imagined speech.

We thank the reviewer for pointing out that this is not described well enough. In our approach, the system continuously decodes the neural activity to generate a continuous acoustic output. There is no explicit speech onset detection or threshold generating sound versus silence. We attribute the effective gating of speech intervals to our discretized spectrogram decoding approach. Particularly, the analysis in Figure 4(D) quantifies the amount of synthesized speech during intervals when the participant was not prompted to generate or imagine speech. We have addressed this comment by clarifying the approach in the *Decoding Approach* section in lines 60-63 by adding:

In this design, an auditory waveform is continuously produced, independent of explicit detection of speech or silence onset. When functioning as intended, the system will provide audible feedback when brain activity patterns representing speech production or imagery are detected and silence otherwise. This way, no additional speech onset detection is necessary.

- the classification approach is not fully clear. What were the class labels to train the LDA? This section needs some more sentences so that a reader can better understand what you exactly did.

We sincerely apologize that we didn't describe the classification approach well enough. In our approach, we first discretize the spectral power in each frequency bin into 9 discrete levels (low power in the spectral bin to high power in the spectral bin). The LDA classifiers then predict the power level based on the neural features. We tried to make this approach clearer by reformulating the corresponding passage in lines 74 – 80 in the section *Decoding Approach* to:

The decoding models assign vectors of neural activity to classes of discretized audio spectrograms. For this purpose, the audio spectrogram is mel-scaled $\text{\cite{stevens1937scale}}$ using 40 triangular filter-banks. In each spectral bin, the signal energy is discretized into 9 energy levels based on the sigmoid function $\text{\cite{angrick2020speech}}$. This quantization enables a high resolution in the spectral range that contains the participant's voice but also provides sufficient information for reliably representing silence. Each of these 9 energy levels represent one of the target classes in the classification approach. Regularized LDA-classifiers then predict the energy level for each of these spectral bins. Thus, the system consists of 40 LDA-classifiers that are trained on the open-loop data from the first experimental run.

And in lines 388-393 in the *Methods* section to:

Here, we focus on modelling strategies that decode waveforms with a constant volume to avoid such unnatural spikes in loudness. Therefore, we expressed the regression task as a classification problem: rather than establishing a direct mapping, we discretized the continuous feature space of each

spectral bin into a fixed set of 9 energy levels. Individual classifiers were then used to predict the energy levels in each individual spectral bin. To recover the speech spectrogram, we applied a dequantization step that replaces all indices with proper spectral coefficients represented by each interval.

- Figure 3B: statistics for each frequency bin: did you account for multiple testing?

We did correct for multiple testing (Bonferroni correction), but completely failed to mention so. We thank the reviewer for the thorough evaluation. This has now been added to lines 105-106:

The proposed method significantly outperforms the baseline in every frequency bin (Mann-Whitney U test, $p < 0.001$, Bonferroni corrected).

When checking these statistics again, we also realized that in Fig. 3 (B) we plotted the distribution of means, instead of plotting the full distribution across all folds for the chance level. While this doesn't change our results, we still corrected the illustration of the chance level in Fig. 3 (B).

- did you always give the words to be spoken in advance? did you try to let the participant imagine words that you did not know in advance?

The participants were always instructed which word to read next and we did not investigate spontaneous speech production. This is an important next step. We have addressed this in lines 209-210 in the *Discussion* by adding:

Additionally, we only investigated the production of individual, prompted words. In the future, spontaneous speech processes need to be investigated to move toward natural conversation.

- The results are impressive, however, at least for a non-dutch listener hard to assess. You could let dutch people listen to the synthesized words and let them rate what they hear. This would give another picture to a reader, how good the quality of the speech recognition and synthesis is.

We absolutely agree with the reviewer that listening tests by native speakers are the ultimate and most useful evaluation of the synthesized output. However, we fully acknowledge that the synthesized speech for this first attempt at real-time imagined feedback is unintelligible and thus would be futile to further confirm this with listening tests. We have mentioned so in the conclusion and results, but agree that it can be further emphasized earlier in the manuscript. We have included the following in the *Abstract*:

While reconstructed audio is not yet intelligible, our real-time synthesis approach represents an essential step towards investigating how patients will learn to operate a closed-loop speech neuroprosthesis based on imagined speech.

Reviewer #2

The authors claim on the basis of their findings “that speech production and imagined speech production share a common neural substrate” (lines 168-169). However, in the neurolinguistic literature on speech production and speech monitoring (see e.g. Christoffels et al., 2007, HBM; also Christoffels et al., 2011, PLoS One) overt and covert (i.e. imagined) speech production lead to huge differences in brain activity, not only in motor-related areas. Interestingly, the current paper claims that the neural signal for speech planning (high gamma band activity) recorded from the electrodes in the three different conditions was comparable or similar enough for the decoder – trained on overt speech – to extract the necessary information in the whispered and imagined speech conditions. I believe it is worth mentioning this discrepancy and explain why this discrepancy exists.

We agree with the reviewer that our claim was overly simplistic and much too brief to make our point. We expanded this to convey what we meant and included the excellent suggested references. We therefore added to lines 176-181 in the *Anatomical and Temporal Contributions* section:

These activation patterns trained on the open-loop run with audible speech also consistently decode speech from the whispered and imagined speaking modes, which point to a role in speech planning and execution as opposed to the perception of the participant's own voice. Differences in neural activation patterns have been reported for different acoustic and proprioceptive feedback conditions \cite{christoffels2007neural,christoffels2011sensory}, but our decoding approach still identifies patterns that are similar enough across the three speaking modes to generate consistently gated synthesized output. This further implies that speech production and imagined speech production share a common neural substrate to some extent, which has also been proposed previously \cite{oppenheim2010motor}.

And further in lines 189-193 in the *Discussion*:

Our decoding models rely predominately on high-gamma features from the frontal cortex and motor areas, which have been implicated in existing models of speech production \cite{guenther2006neural,tourville2011diva,hickok2012computational}.

This also offers an explanation as to transferability of the decoding models to whispered and imagined speech. While these three processes are clearly different in terms of acoustic and proprioceptive feedback \cite{christoffels2011sensory}, our results indicate that they share enough neural activity to enable model transfer. The selected area can also function as a blueprint for future implants in patient cohorts.

Regarding their methodology, the authors need to make clear what the difference between closed- and open-loop experiments/conditions is. In the overt speech production condition, the patient read aloud the words on the screen and heard herself (via the internal-cognitive as well as external – acoustic and bone transmission – monitoring loop). In the whispered condition, this is comparable, only that the acoustic signal is much weaker and hence also the bone transmission. However, it is not clear to me why the overt condition is labeled as “open loop” and the whispered condition as “closed loop”. Finally, in the imagined naming condition, the patient can only monitor herself via the internal monitoring loop (since no motor movements are carried out and no

acoustic output is produced). Therefore, the imagined speech condition may be a closed loop in the sense that there is no external monitoring for the patient. The difference between open- and closed-loop experiments should be made explicit to avoid confusion.

Re-reading the manuscript, we now realize that the employed terms are only understandable for a very small community and that our description is in fact not clear to the intended audience. We therefore added the following sections in lines 43-46 in the *Intro* to clarify what we mean by the terms open-loop and closed-loop:

In this context, an open-loop experiment refers to the collection of data without immediate feedback provided by the neuroprosthesis. In a closed-loop experiment, the neural data is analyzed and decoded in real-time and feedback is provided by the neuroprosthesis. The neuroprosthesis is thereby closing the loop.

It is well established, by the way, that when speakers monitor their output even when they do not produce any sounds. This is called internal monitoring (see e.g. Levelt et al., 1999, BBS). A potential problem with the imagined speech condition is that the authors cannot be sure that the patient cooperated and how well she imagined producing the words covertly.

As mentioned above, the “reconstructed speech is not intelligible” (line 88). Given this fact, the authors need to state the purpose of the feedback, i.e. why was the reconstructed speech played back to the patient if it was incomprehensible anyway? I could imagine that once the signal can be improved feedback would fulfil a function, however, under the current conditions, the purpose is not clear to me.

We thank the reviewer for this helpful comment. In our experience, the compliance of the participant is in fact the crucial point in imagined speech experiments. The experimenters never know if the participant really is imagining to produce speech and if so, when it started, and how long the participant required to produce the imagined speech. Because of this, we believe it is fundamentally important to investigate neuroprosthesis based on imagined speech processes with direct feedback provided by the decoding approach. This way, the participant immediately receives feedback which type of processes are decoded by the neuroprosthesis and which processes are not. Additionally, this way the experimenters do see whether the participant engaged in the task. We therefore added lines 213-216 to the *Discussion*:

By synthesizing imagined speech processes for the first time, we are able to actively engage the participant in the imagined speech experiment, effectively allowing the participant to monitor the speech production process and adjust accordingly \cite{christoffels2007neural}. This process will be critical as the user learns to use and control the neuroprosthetic, and eventually for co-adaptation between the user and system. These initial results provide evidence that neural activity during speech imagery can be used in the development of future speech neuroprosthesis.

Kind Regards,

Dr. Christian Herff
Assistant Professor

Reviewers' comments:

Referee #1:

The authors addressed all my concerns and added important new information to the manuscript.

Referee #2:

This is the revision of a manuscript that I reviewed before. I believe that the manuscript has further improved. The authors answered my questions carefully. There are only small issues that they should attend to before the manuscript is ready to be published.

My first major point is answered to my satisfaction. My second point, a methodological issue (closed- vs. open-loop experiments), has been answered as well. Although their answer is relatively concise, I believe that it is sufficient. However, I wonder why the authors call their experiments open- and closed-loop (with the potential confusion these terms may cause) – why not simply say experiments with and without feedback? To me, this would have been much clearer in the first place.

In response to my third major point (intelligibility of the reconstructed speech), the authors wrote “Because of this, we believe it is fundamentally important to investigate neuroprosthesis based on imagined speech processes with direct feedback provided by the decoding approach. This way, the participant immediately receives feedback which type of processes are decoded by the neuroprosthesis and which processes are not”.

I have to admit that I do not understand completely what the authors mean by saying that the participant immediately receives feedback about the type of stimulus when they stated earlier that the decoded speech is unintelligible. For instance, the participant imagines to produce the word on the screen (covert production), generates brain activity which is picked up and decoded, and the resulting signal – which is not intelligible – is feedback to the participant. Why? (if it's not intelligible) Probably I miss something here, however, maybe the authors should try and clarify again.

Referee #2: This is the revision of a manuscript that I reviewed before. I believe that the manuscript has further improved. The authors answered my questions carefully. There are only small issues that they should attend to before the manuscript is ready to be published. My first major point is answered to my satisfaction. My second point, a methodological issue (closed- vs. open-loop experiments), has been answered as well. Although their answer is relatively concise, I believe that it is sufficient. However, I wonder why the authors call their experiments open- and closed-loop (with the potential confusion these terms may cause) – why not simply say experiments with and without feedback? To me, this would have been much clearer in the first place.

We used the terms open- and closed-loop as they are widely established in BCI and, in particular, neuroprostheses communities, see for example [1-4]. We hope that with the explanation added in the last revision, the terminology is clear to the audience outside of the neuroprostheses community, as well.

[1] Chase, S. M., Schwartz, A. B., & Kass, R. E. (2009). Bias, optimal linear estimation, and the differences between open-loop simulation and closed-loop performance of spiking-based brain-computer interface algorithms. *Neural networks*, 22(9), 1203-1213.

[2] Koyama, S., Chase, S. M., Whitford, A. S., Velliste, M., Schwartz, A. B., & Kass, R. E. (2010). Comparison of brain-computer interface decoding algorithms in open-loop and closed-loop control. *Journal of computational neuroscience*, 29(1-2), 73-87.

[3] Hochberg, L. R., Bacher, D., Jarosiewicz, B., Masse, N. Y., Simeral, J. D., Vogel, J., ... & Donoghue, J. P. (2012). Reach and grasp by people with tetraplegia using a neurally controlled robotic arm. *Nature*, 485(7398), 372-375.

[4] Orsborn, A. L., Moorman, H. G., Overduin, S. A., Shانهchi, M. M., Dimitrov, D. F., & Carmena, J. M. (2014). Closed-loop decoder adaptation shapes neural plasticity for skillful neuroprosthetic control. *Neuron*, 82(6), 1380-1393.

In response to my third major point (intelligibility of the reconstructed speech), the authors wrote “Because of this, we believe it is fundamentally important to investigate neuroprosthesis based on imagined speech processes with direct feedback provided by the decoding approach. This way, the participant immediately receives feedback which type of processes are decoded by the neuroprosthesis and which processes are not”. I have to admit that I do not understand completely what the authors mean by saying that the participant immediately receives feedback about the type of stimulus when they stated earlier that the decoded speech is unintelligible. For instance, the participant imagines to produce the word on the screen (covert production), generates brain activity which is picked up and decoded, and the resulting signal – which is not intelligible – is feedback to the participant. Why? (if it’s not intelligible) Probably I miss something here, however, maybe the authors should try and clarify again.

We believe that direct continuous feedback is crucial for targeting imagined speech processes, even though the produced speech is not intelligible at this point. As mentioned by Reviewer 2 in the initial comment, there is always a potential uncertainty about the extent to which the participant follows the instructions of the imagined speech condition. The purpose of the feedback addresses this problem and gives control to the participant for enabling an interaction with the decoder, contrary to open-loop experiments in which it is difficult to assess whether the participant followed the instructions of the imagined speech task. Therefore, even though the feedback is not intelligible at this point, it still fulfills important functions: On one hand, the participant is aware that she is in control of the system and has the opportunity to adapt her speech imagery accordingly to the perceived output. On the other hand, the immediate feedback might provide the ability to practice

generating better and better speech output, when longer experiments are conducted in the future. We made this argument clearer by adding in lines 209-210 of the discussion:

The immediate feedback provided by our system will allow patients to learn to operate the prosthesis and improve synthesis quality gradually.